# Establishing Animal Welfare Rules of Conduct for the Portuguese Veterinary Profession—Results from a Policy Delphi with Vignettes

**DOI:** 10.3390/ani10091596

**Published:** 2020-09-08

**Authors:** Manuel Magalhães-Sant’Ana, Maria Conceição Peleteiro, George Stilwell

**Affiliations:** 1CIISA—Centre for Interdisciplinary Research in Animal Health, Faculty of Veterinary Medicine, University of Lisbon, 1300-477 Lisboa, Portugal; mcpelet@fmv.ulisboa.pt (M.C.P.); stilwell@fmv.ulisboa.pt (G.S.); 2Ordem dos Médicos Veterinários, Av. Filipe Folque, 10J, 4º Dto., 1050-113 Lisboa, Portugal

**Keywords:** animal law, animal welfare, code of conduct, euthanasia, emergency slaughter, fitness for transport, Policy Delphi, veterinary ethics, vignettes, moral agency

## Abstract

**Simple Summary:**

Promoting animal welfare is one of the basic tenets of the veterinary profession. However, it is known that the Portuguese veterinary code of conduct, established more than 25 years ago, overlooks animal welfare and fails to address issues such as the euthanasia and humane killing of animals. Using a Policy Delphi debate technique with vignettes, this study assesses the level of agreement of forty-one (out of seventy) purposively selected Portuguese veterinarians on ten animal welfare requirements set forth in a code of professional conduct, meant to be approved in a near future. Most participants agreed that the suggested animal welfare rules of conduct reflected their own views on the subject (88%), in addition to representing a significant improvement in terms of regulatory standards (93%). This study will support regulation and policy-making by the Portuguese Veterinary Order and by veterinary representative bodies elsewhere.

**Abstract:**

Promoting animal welfare is one of the basic tenets of the veterinary profession and, in doing so, veterinarians are expected to abide to the highest legal and professional standards. However, the Portuguese veterinary code of conduct, established in 1994, largely overlooks animal welfare and fails to address issues such as the euthanasia or humane killing of animals. As part of a wider research aiming to revise the Portuguese veterinary code of conduct, a Policy Delphi study was conducted in late 2018, using a pre-validated three-round structure and vignette methodology, to explore the range of opinions and the level of agreement on end-of-life dilemmas and animal welfare rules of conduct of a purposeful sample of forty-one (out of seventy) Portuguese veterinarians. When faced with ethical vignettes involving end-of-life dilemmas, veterinarians were shown to privilege personal moral agency over legal obligations in order to defend the interests of stakeholders, namely of the animals. Most participants agreed that the suggested animal welfare rules of conduct reflected their own views on the subject (88%), in addition to representing a significant improvement in terms of regulatory standards (93%). We expect that this study will support regulation and policy-making by the Portuguese Veterinary Order and by veterinary representative bodies elsewhere.

## 1. Introduction

Promoting animal welfare is one of the basic tenets of the veterinary profession. Making veterinarians aware of their duties towards animals can be achieved in several ways, namely through academic education or by instituting professional standards. However, while the former has received considerable attention by the veterinary literature, and recent evidence suggests that the teaching of animal welfare science, ethics, and law has increased amongst European veterinary schools [1], less is known about the latter.

Few studies have investigated which provisions are presently in place to promote animal welfare in veterinary professional standards, namely in codes of professional conduct. A systematic review found substantial differences in terms of animal welfare requirements set out in five European veterinary codes of professional conduct, namely from Ireland, UK, Denmark, Portugal, and the Federation of Veterinarians of Europe [2]. In particular, the Portuguese Veterinary Code of Conduct (*Código Deontológico da Ordem dos Médicos Veterinários*—CD-OMV) was found to be mainly focused on regulating the relationships between veterinary professionals, while neglecting duties towards animals [2]. For example, it was identified that the CD-OMV overlooks animal welfare and fails to address issues such as the euthanasia or humane killing of animals. This review also suggested that the CD-OMV has not been adapted to accommodate societal changes regarding the moral standing of animals [2].

End-of-life decisions are arguably amongst the most challenging ethical dilemmas faced by veterinarians [3]. Ethical dilemmas involve the conundrum of attending simultaneously to two (or more) morally relevant and competing interests [4]. In their daily practice, veterinarians are expected to manage the difficult balance between quality and quantity of animal life, two welfare dimensions that often conflict [5]. At the very least, veterinarians are bound to provide animals at their care with a good death, irrespective of their beliefs on the moral standing of animals. Two prominent examples of ethical dilemmas with animal welfare implications are the euthanasia of stray animals by local veterinary officers and the humane killing of farm animals by veterinary practitioners.

In Portugal, a recent legal document (Law no. 27/2016) has restricted the euthanasia of unowned companion animals to “cases of proven incurable disease and when it proves to be the only and indispensable means to eliminate irretrievable pain and suffering” [6]. Overpopulation, financial constraints, or owner’s convenience are no longer considered acceptable reasons for euthanising animals. The law, promoted by some Members of Parliament as a breakthrough in animal protection legislation, was received with scepticism by the veterinary community because there was concerns that its enforcement would increase rather than decrease stray animal overpopulation. Moreover, the law has constrained the autonomous professional judgement of local veterinary officers, thus contributing to the complex conflict between personal ethical beliefs and legal requirements that characterises the daily job of official veterinarians [7].

With regard to farm animal practice, significant animal welfare concerns arise from emergency and casualty slaughter veterinary certification [8,9,10,11]. An analysis of disciplinary proceedings against veterinarians in Portugal revealed that 18% of complaints concerned breaches of animal welfare protection [12], particularly when issuing certificates for transporting animals to the slaughterhouse that allegedly should have been subjected to on-farm emergency slaughter. In such cases, veterinarians are faced with the ethical dilemma of attending primarily to the welfare of the animal or to the interests of the farmer [10,13]. European Council Regulation (EC) no. 1/2005 requires that animals “*that are unable to move independently without pain or to walk unassisted*” should be considered unfit for transport and treated or slaughtered on-farm [14]. In order to help decision-making by assistant veterinarians, the Portuguese Veterinary Authority (*Direção Geral de Alimentação e Veterinária*—DGAV) issued a Good Practice Guide for Fitness for Transport and Emergency Slaughter in 2012 [15], but lack of means (both human and material) for on-farm emergency slaughter has often been invoked to justify transporting otherwise unfit animals (especially adult cattle) [12].

Since September 2017, the Portuguese Veterinary Order (*Ordem dos Médicos Veterinários*—OMV) has supported a research project aimed at gathering evidence to inform the development of a new code of professional conduct. The CD-OMV has remained virtually unchanged since its inception in 1994 [2] and its origins can be traced back to 1979, to a code of conduct developed by the Portuguese veterinary trade union (*Sindicato Nacional dos Médicos Veterinários*—SNMV). Against this background that requires updating and improving current veterinary professional standards, the aim of this study is to explore the range of opinions and the level of agreement on animal welfare requirements (with an emphasis on end-of-life decisions) of a purposeful sample of Portuguese veterinarians, using a Policy Delphi technique with vignette methodology.

## 2. Materials and Methods

As part of a wider investigation aiming to revise the CD-OMV, a Policy Delphi study was conducted between 10 September–14 December 2018, using a pre-validated three-round structure and vignette methodology [16,17]. Vignettes have been described as “short stories about hypothetical characters in specified circumstances, to whose situation the interviewee is invited to respond” [18] (p. 105). A Policy Delphi is a structured group-facilitation technique in which a restricted group of experts—in this case a purposeful sample of veterinarians that may represent the veterinary profession in Portugal—remotely discuss complex topics through several rounds, under a blanket of quasi-anonymity (i.e., their identity is only known to the researcher facilitating the Delphi) [19]. The facilitator is responsible for introducing the research topics, collecting participants’ responses, and producing a summary of responses in order to inform the next round of questions.

The Policy Delphi was applied here to investigate three topics identified in a retrospective analysis of disciplinary proceeding set forth against veterinarians in Portugal [12], namely advertising, telemedicine, and animal welfare. Findings regarding telemedicine have been described elsewhere [20]. Overall results from the Policy Delphi will be incorporated into new professional guidance (revised CD-OMV) and submitted for consultation amongst the Portuguese veterinary community before a final document is produced (Figure 1).

Round 1 of the Delphi dealt with advertising and telemedicine (data not shown). In Round 2, participants were briefed about the research topics (Appendix B, in Portuguese) and presented with two vignettes, roughly 100 words long, describing ethical dilemmas involving animal welfare: One on euthanasia at an animal shelter and another on fitness for transport and on-farm slaughter (Table 1). The vignettes were inspired by real-life cases emerging from previous research [12]. Using a 5-point Likert scale (1—strongly disagree; 5—strongly agree), participants had to rate a set of statements pertaining to the vignettes and asked to justify at least two answers. Participants were also asked to explore how the CD-OMV could help veterinarians dealing with similar situations in the future (open-ended questions).

In Round 3, participants were provided with the level of agreement achieved in the previous rounds and with a summary of the arguments that were used. In terms of animal welfare, ten rules of conduct, to be included in the revised CD-OMV, were submitted to the appreciation of participants. These rules of conduct were written based on the responses to Round 2. Participants were asked about their level of agreement and invited to suggest changes or alternatives. Another set of questions addressed the structuring of the future CD-OMV (data not shown). A “no answer” (N/A) option was always available.

Each round (including invitation) was piloted by five senior veterinarians, not involved with the study, and with proven professional experience, including research, regulation, education, clinical practice, and surgery. In terms of study population, the selection criteria aimed for diversity and included:Veterinarians registered with the OMV and working in Portugal;No previous sanctions within the last five years or outstanding fees;Policy-making experience with Portuguese veterinary organisations;Gender balance;Proportionality in terms of age groups;Broad geographical distribution in terms of working location;Broad professional experience and field of work;Diverse educational backgrounds.

Snowball sampling was used for recruitment and seventy individuals were identified and invited to participate by email via SurveyMonkey. On invitation, prospective panelists were informed about the aims and methods of the study, including that their identity would remain anonymous, and a written consent was sought. Participants were identified by a four-digit alphanumeric code (e.g., sp25) after submitting their demographic profile. In order to yield more meaningful results, and avoid acquiescence bias, participants were instructed to respond according to their personal beliefs and not according to current rules and standards. Quantitative material was handled using Microsoft Excel version 1908. For description purposes, responses for strongly disagree/disagree and for agree/strongly agree were aggregated. Overall results from data analysis, including means and standard deviations (mean ± SD), are provided as Appendix A. Qualitative material was introduced in NVIVO (QSR International 2020) and analysed by content, following the preparation, organizing, and resulting phases described by Elo and Kyngäs [21]. A visual representation of the generated codes is provided as Appendix A. Quotes were translated to English by M.M.S. and, when necessary, edited to facilitate readership while maintaining their original meaning. The study conforms to ethical standards approved by *Conselho Profissional e Deontológico* (Ethics Council), *Ordem dos Médicos Veterinários* (reference number: 673/CPD/2017).

## 3. Results

### 3.1. Demographics

Of the seventy invitations, forty-one veterinarians representing the diversity of the veterinary profession in Portugal accepted to participate (59% acceptance rate), and all participants responded to the three rounds (100% response rate, including N/A). Respondents’ demographic profile has been described before [20] and is summarised herein: 61% were male, 70.8% had a veterinary degree/master as highest qualification, 56.1% were under the age of 45, and 51.2% worked in companion animals (all main areas of veterinary activity were represented, with 49% indicating more than one area). In terms of participants’ working location, all districts of continental Portugal were represented, except two, as well as Madeira and Azores Archipelagos, and this geographical distribution roughly mirrored the overall population distribution [20].

In addition, 83% of respondents had current or previous policy-making experience with nineteen Portuguese veterinary organisations, including the representative body (OMV), the national veterinary authority (DGAV), the trade union (SNMV) and professional organisations for companion animals (APMVEAC), ruminants (APB), equine (APMVE), local veterinary officers (ANVETEM), pig science (SCS), veterinary research (SPCV, INIAV), alternative therapies (APAMV), etc. (Figure 2).

### 3.2. Vignette: Euthanasia

Most participants considered that, by deciding to euthanase the cat, the veterinarian’s conduct was justified under the circumstances (thirty-two, 78%), having taken into account the best interests of the public (twenty-nine, 71%) as well as those of the animal (twenty-eight, 68%). Thirty-two participants (78%) thought that the vet took the best possible decision, a decision that should not constitute a disciplinary offense (thirty-two, 73%) nor should it be punished by the veterinary regulator, the OMV (thirty-five, 85%). In addition, the majority of panelists did not agree that the cat should have been handed over to an animal charity (twenty-four, 58%). A visual representation of results can be found in Figure 3.

In their comments, respondents expressed their discontent with the current legal framework, enforced regardless of the lack of effective measures to tackle the relinquishment and abandonment of companion animals, and of how it has made the work of local veterinary officers increasingly difficult. Competing opinions arose regarding the scope of the law. One veterinary officer (xv60) argued that restrictions to euthanasia also apply to private veterinary practitioners, although an exotic veterinary practitioner (sp34) disagreed by saying that “*it is only strange that, faced with the same clinical case, a vet cannot resort to euthanasia if the animal is stray, but may do the opposite if the owner of the animal so wishes and asks the vet*”.

Participants highlighted the need to allow vets to take autonomous value-based decisions and justified their support for euthanasia mostly because of animal welfare concerns, followed by public health. They also mentioned that the code of conduct needs to consider how individual clinical and ethical judgments must precede, or even supersede, legal impositions, because, in the words of a farm animal practitioner (bv51), “*legitimate illegalities supersede illegitimate laws*”. Other panelists thought that the code should take a more prescriptive approach to animal euthanasia. A small animal practitioner (sp31) suggested that “*the code of conduct should unequivocally include criteria that would allow the veterinarian to make the decision to euthanase consciously and calmly, minimizing any doubts that might arise in face of situations [such as these]*”. Another practitioner (sp70) went so far as to suggest that the CD-OMV should allow animal euthanasia when the prognosis is poor or suffering is involved. The impact of economic constraints in decision-making was a recurrent topic, an impact that the future code of conduct should take on board, although a local veterinary officer (xv15) cautioned that “*lack of resources cannot be used to justify non-intervention, when it would be up to the [veterinary] professional to give the alert to fill this lack*”.

### 3.3. Vignette: Transport and On-Farm Slaughter

Two-thirds of participants considered that, under the circumstances, the vet’s conduct is justified (twenty-seven, 66%). Participants clearly agreed that the vet acted in the best interest of the farmer (thirty-eight, 93%) but there was little consensus on whether the interests of the public (37% neutral) and of the animal (44% broadly agree, 30% broadly disagree) were also met. A small majority of participants (twenty-two, 54%) considered that the vet took the best possible decision. More than half disagreed that the vet’s conduct should be considered a disciplinary offense (twenty-one 51%) or that the practitioner must be punished by the OMV (twenty-five, 61%). Detailed results can be found in. Figure 4.

In their comments, participants emphasised the conflict between the legal framework and real-life decisions. Most of the arguments dealt with minimizing animal suffering, while safeguarding public health concerns and the well-being of the farmer. In defence of the vet’s conduct on animal welfare grounds, an equine veterinarian (as05) reasoned that “*if no such decision was taken, the farmer would have slaughtered the cow himself, irrespective of necessary knowledge and skills to do so (or worse: the cow would be left to die in suffering), (…) the vet’s intervention (…) safeguarded the best interest of the animal, even if indirectly*”. A similar line of reasoning was made by an experienced farm animal veterinarian (lp37):
“*Unfortunately, we are often forced to deal with this dilemma (…) objectively, slaughter outside the slaughterhouse *[on-farm emergency slaughter]* is not an option. There are often no means available and when they do exist, it takes much longer than having the cow slaughtered at the slaughterhouse. That is, if we defend on-farm slaughter, (...) welfare is not improved since the animal is often subjected to many more hours/days of suffering*”.

Other respondents disagreed, namely official veterinarians. For example, one official veterinarian (xv01) argued that “*any assistant veterinarian must be knowledgeable of the permitted on-farm methods of slaughter and be prepared with the necessary means to perform it*”, whereas another (xv13) considered that euthanasia would have been preferable. Some participants argued that more information was needed in order to decide, namely concerning the specifics regarding the cow’s clinical condition. As in the previous vignette, the topic of insufficient available resources emerged frequently, namely denouncing lack of coordinated efforts between stakeholders to effect on-farm emergency slaughter. A local veterinary officer (xv12) suggested that:
There should be an interconnection of services, namely between assistant veterinarians, local veterinary officers and DGAV officials, to gather the necessary equipment that would allow direct collaboration to resolve cases of slaughter outside the slaughterhouse.

### 3.4. Revised Guidance

In Round 3, participants were presented with ten rules of conduct describing the duties of veterinarians towards animal welfare, to be included in the revised CD-OMV, and partially inspired by results from Round 2, as follows:Veterinarians must be respectful of animals, avoiding violence and unnecessary suffering in their handing, restraint, treatment or transport.Veterinarians must be aware of animal health and welfare legislation.Veterinarians must report to the competent authorities cases that may result in unjustified suffering, abandonment or death (safeguarding professional secrecy, where applicable).Veterinarians must provide first aid to animals, according to their competence.Likewise, it is the duty of any veterinarian to ensure that animals in irretrievable suffering are euthanased or slaughtered as soon as possible and using methods deemed fit for purpose.Only veterinarians may decide and practice animal euthanasia.The decision to euthanize an animal shall consider, in addition to animal health and welfare, public health as well as the legitimate interests of its owner or keeper.The preceding points do not exclude the delegation of veterinary acts in urgent cases, epidemics or disasters.The veterinarian is required to obtain consent from the rightful owner or keeper of the animal prior to any treatment or euthanasia.The cases described in points 4 and 5 do not require consent, although it should be sought prior to practice.

Thirty-eight participants (93%) broadly agreed that these new rules substantially improve animal welfare protection and thirty-six (89%) considered that they reflect their own views regarding animal welfare. Nonetheless, twenty-eight participants (69%) broadly agreed that improvements are still required (Figure 5).

Participants concurred that the future code of conduct should assist with decision-making, but two different approaches emerged: Most suggested that the CD-OMV should be more prescriptive, clearly defining and detailing the accepted limits of practice when animal welfare is involved; others, however, considered that the code of conduct should provide guidance in the form of algorithms (sp24) or decision trees (xv49). With regard to end-of-life issues, these could include quality of life assessment protocols (sp62) or critical points in the euthanasia decision process, such as zoonoses (sp34) or poor prognosis (sp24). Another suggestion included how the code should be dynamic and adapted to societal and legal changes (lp64). Finally, it was also suggested that veterinarians should be required to keep a record of euthanasia, including which circumstances led to those decisions (lp47), and the establishment of a support line (lp40), to help veterinarians dealing with urgent cases.

## 4. Discussion

This study used a Policy Delphi with vignette methodology to explore prominent end-of-life topics and inform the development of animal welfare requirements for the new Portuguese Veterinary Code of Conduct. A considerable amount of literature endorses the use of the Delphi technique to investigate socially acute questions [19,22]. In particular, the Policy Delphi has proven to be an effective method to explore the level of agreement on difficult or unexplored topics because it does not necessarily require reaching a final consensus [19]. The state of quasi-anonymity ensures confidentiality and enables participants to safely express their opinions, without the partisanship that often hinder face-to-face discussions. In turn, incorporating vignettes in Delphi studies has been considered useful for empirical ethics [23], namely to investigate such topics as horse welfare [17] and veterinary ethics education [24]. Vignettes used in this research described genuine ethical dilemmas, in which an ethically correct course of action was not easily discernible. Moreover, the vignettes alluded to the three dimensions of animal welfare: Scientific knowledge, ethical decision-making, and the relevant legal framework to elicit meaningful competitive arguments on the research topic.

Significantly, the veterinarians’ conduct in both Round 2 case scenarios was considered within acceptable limits, even though they were potentially in breach of legal requirements. When faced with an ethical dilemma involving animal welfare, participant veterinarians weighed the interests of animals against those of other relevant stakeholders. Instead of privileging animals’ inherent value and *prima-facie* moral duties, participants approached both ethical dilemmas from a utilitarian perspective, mostly trying to maximize gains and minimise costs for all those involved. Most notably, they were shown to favour personal moral agency over legal obligations in order to defend the interest of stakeholders, including—but not limited to—animals.

The first vignette illustrated an ethical dilemma where a local veterinary officer was presented with a stray cat, with a potentially curable zoonosis (dermatophytosis), and suffering from traumatic and painful lesions requiring surgical treatment (also potentially treatable). Resorting to euthanasia fell outside the letter of Law no. 27/2016, insofar the cat did not suffer from an obviously incurable disease, nor had been confirmed that euthanasia would be “the only and indispensable means to eliminate irretrievable pain and suffering” [6]. However, the cat was feral, and subjecting it to surgery and prolonged dermatological treatments could be considered objectionable from a welfare point of view. At least for most Delphi participants, prolonging animal life was antithetical to good welfare. Participants acknowledged that the vet’s decision accounted for the best interests of the animal as much as protected public health, and that it should not constitute a disciplinary offense.

In Portugal, the euthanasia of healthy animals in public shelters has been prohibited since 2016 (Law no. 27/2016). A later ordinance (Portaria 146/2017, 26 April) clarified that zoonoses or other infectious diseases can be considered acceptable reasons for euthanasing animals “*if their permanence in the shelter represents a threat to animal health, or a danger to public health, within or following an infectious disease outbreak*” [25]. This has created an overlap between competing regulatory documents (e.g., could an animal suffering from a zoonosis, but not part of an outbreak nor proven incurable, be euthanised?), a confusion that was reflected in participants’ responses. It is also not entirely clear whether the provisions put forth in these laws also include private veterinary practices and whether restrictions apply to animal euthanasia in clinical context other than shelters. These questions call for an urgent revision of the current regulatory framework, a task that a recently established “Animal Welfare Working Group” is expected to tackle [26]. They also present additional challenges for developing meaningful veterinary professional standards.

The restrictions imposed to local veterinary officers in Portugal on the euthanasia of unowned companion animals are burdensome, but they are not unique. An Austrian study described how “*working as a veterinary officer often means searching for a ‘best possible’ in a ‘worst case’*” and that working with conflicting norms is the rule rather than the exception [27]. Other countries, such as Italy, also restrict euthanasia to dangerous or incurably ill animals (Italian Framework Law no. 281/1991 [28]). However, Italian veterinarians are not expected to demonstrate that euthanasia is the only means to eliminate irretrievable pain and suffering, a request that may well be impossible to determine in most cases.

Regarding the second vignette, the veterinary clinician is confronted with a downer cow, unable to rise without help. In the absence of the necessary means to carry out on-farm emergency slaughter (due to the lack of emergency services by the slaughterhouse, the lack of conditions to appropriately bleed/eviscerate the animal, or because there is no captive bolt pistol available), the veterinarian is faced with the ethical dilemma of attending primarily to the welfare of the animal (and slaughter it on-site) or to the interests of the farmer (and having the animal transported to the slaughterhouse as soon as possible, in breach of European Regulation no. 1/2005).

Results indicate a higher level of uncertainty than in the previous vignette. Participants were unsure on whether the interests of the cow and public health were also attended by the vet’s decision. The dilemma is tangible: On the one hand seeking medical treatment could have caused prolonged and unnecessary suffering and, on the other hand, the on-site stunning, hoisting, bleeding, handing, and transport of the carcass would have caused public health concerns. A third option—euthanasia—would have caused a total loss for the farmer. The solution to this dilemma may be differently perceived by veterinary practitioners and by official veterinarians [8,10], a conflict seemingly corroborated by our findings.

Too often, the certification of fitness for transport or the alternative decision to slaughter farm animals on site may have severely compromised animal welfare as studies from Ireland [9], Denmark [29], and Canada [30] seem to suggest. These studies also show that Portuguese veterinarians are not alone in their reservations on how to deal with cases of fitness for transport, thus claiming for a revision of current policies. A report from the ONG Animals’ Angels has exposed the lack of enforcement of Regulation no. 1/2005 throughout Europe and particularly in Portugal [31], where at least a few cases have resulted in disciplinary proceedings against veterinarians for the certification of an illegal act [12].

Everything considered, professional standards need to take on board the specific challenges faced by veterinarians when taking end-of-life decisions. Participants in this study stressed the need for having professional standards that promote personal autonomous decisions and respect veterinarians’ moral agency. Including provisions in the CD-OMV for the euthanasia or slaughter of animals in irretrievable suffering (point 5 of revised guidance) empowering veterinarians to act primarily in the best interest of the animal, an act whose decision should not be delegated to non-vets (point 6). On the other hand, by clarifying that these decisions should also consider, in addition to animal health and welfare, public health as well as the legitimate interests of the owner (point 7), promotes decision-making and accommodates personal moral judgments (i.e., moral agency). These principles do not supersede the duty of every veterinarian in being aware of the most recent animal health and welfare legislation (point 2), of providing animals with first aid according to their competence (point 4), and to actively report animal abuse (point 3). Considering that participants represent the diversity of the Portuguese veterinary community, it can be concluded that the ten rules of conduct suggested in Round 3 accommodate the spectrum of opinions regarding animal welfare, in addition to representing a significant improvement in terms of veterinary regulatory standards.

The ten rules of conduct improve, but do not resolve, the needs in terms of animal welfare standards and professional guidance. In effect, over two-thirds of panellists considered that these new rules still require improvement. One area that is currently lacking is that of positive animal welfare. The suggested rules are focused on preventing harm and further efforts are needed to guide Portuguese veterinarians towards promoting positive mental states in animals. Positive welfare is now an integral part of European veterinary curricula [1] but it was probably absent from the education of participant veterinarians.

Furthermore, the revised CD-OMV should empower veterinarians to act as fully-fledged moral agents, i.e., to conscientiously and autonomously devise the best course of action, having considered the available options and the relevant stakeholders (including oneself) [32]. Appeal to moral agency can only be made if veterinarians are aware of the societal debate regarding the moral standing of animals and are competent in ethical decision-making. Educational strategies that promote a pluralistic approach to considering ethical issues involving animal welfare are therefore needed [24,32]. Providing veterinarians with ethical reasoning skills is increasingly relevant given the demands brought by the One Health paradigm. In the words of Nieuwland and Meijboom:
Veterinarians have to operate in a force-field of various responsibilities, obligations, demands and expectations, of which One Health appears only to make more complicated. However, letting complexity and conflicting demands narrow and blunt moral agency should be prevented if only to protect the wellbeing of veterinarians themselves*[33].*

Interestingly, few comments expressed concern with veterinarians’ personal wellbeing, although the format of the Delphi may have partially contributed to this finding (or lack thereof). Faced with a myriad of ethical challenges when confronted with animal welfare issues, it is perhaps unsurprising that veterinarians are especially susceptible to burnout and moral distress [34]. Inability to apply moral agency in everyday professional decisions involving animal welfare can result in moral distress, and a study found that North American veterinarians have little to no training on how to effectively deal with moral stress [35]. Incorporating flowcharts and frameworks into the CD-OMV to assist veterinarians in their ethical decision-making regarding animal welfare can provide such training tools [32].

Some limitations of the present study should be acknowledged. The animal welfare topics were only addressed in two of the three rounds of the Delphi, thus limiting the yielded debate. Since the vignettes were purposively short, not all aspects of the case were detailed or explained, thus allowing participants to fill in the gaps, a task particularly challenging for those less familiar with the research topic. In effect, some participants addressed the concern of insufficient background information and abstained from having an opinion in Round 2 (on average, 4.2% in the first vignette and 5.2% in the second).

Results from this research have laid the foundations for establishing veterinary rules of conduct that may effectively protect animals and improve animal welfare, assist veterinarians working in Portugal with their ethical decision-making, and thus promote their personal moral agency. Further efforts are required to ensure that these rules are fit for purpose, namely by extending the debate to the wider Portuguese veterinary community. The process used to reach these goals may also prove useful for veterinary regulators elsewhere who would like to improve their animal protection standards. Finally, the results and consequent reflection concerning the vignettes may be used by the Portuguese veterinary authority (DGAV) to inform the revision of current animal welfare regulations and enforcement procedures.

## 5. Conclusions

In dealing with end-of-life dilemmas, participant veterinarians were shown to maximise utility for all those involved. The ten rules of conduct suggested by this study provide a starting point to incorporate animal welfare concerns into the Portuguese veterinary code of conduct. An emphasis on positive welfare is still required and additional research is needed to establish how duties towards animals will combine with those towards other stakeholders, namely clients and society. Portuguese veterinarians need professional guidance that promotes moral agency, a competence that requires up to date academic education on the societal views regarding the moral standing of animals.

## Figures and Tables

**Figure 1 animals-10-01596-f001:**
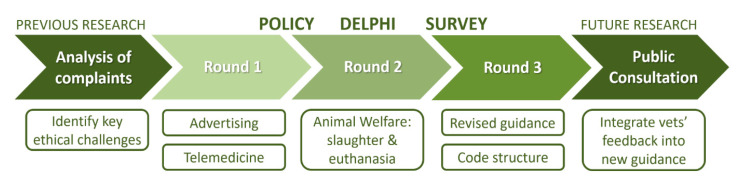
Flowchart describing the topics explored at each of the three rounds of the Policy Delphi and how they relate with previous and future research aiming to yield a new Portuguese veterinary code of conduct.

**Figure 2 animals-10-01596-f002:**
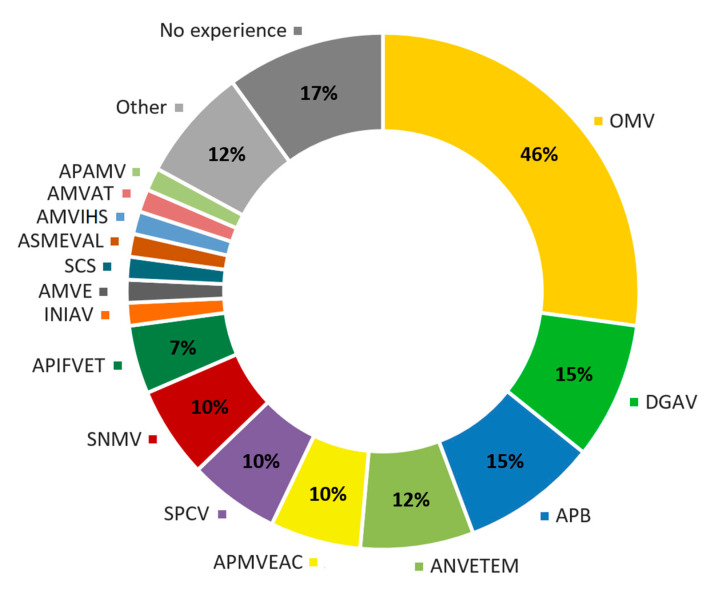
Participants’ policy-making experience (current or previous) with Portuguese veterinary organisations. OMV—*Ordem dos Médicos Veterinários*; DGAV—*Direção Geral de Alimentação e Veterinária*; APB—*Associação Portuguesa de Buiatria*; ANVETEM—*Associação Nacional de Médicos Veterinários dos Municípios*; SPCV—*Sociedade Portuguesa de Ciencias Veterinárias*; SNMV—*Sindicato Nacional dos Médicos Veterinários*; APIFVET—*Associação da Indústria Farmaceutica de Medicamentos Veterinários*; INIAV—*Instituto Nacional de Investigação Agrária e Veterinária*; AMVE—*Associação de Médicos Veterinários de Equinos*; SCS—*Sociedade Científica de Suinicultura*; AMVIHS—*Associação dos Médicos Veterinários Inspectores Higio-Sanitários*; AMVAT—*Associação de Médicos Veterinários com Actividade Taurina*; APAMV—*Associação Portuguesa de Acupuntura Medico-Veterinária*.

**Figure 3 animals-10-01596-f003:**
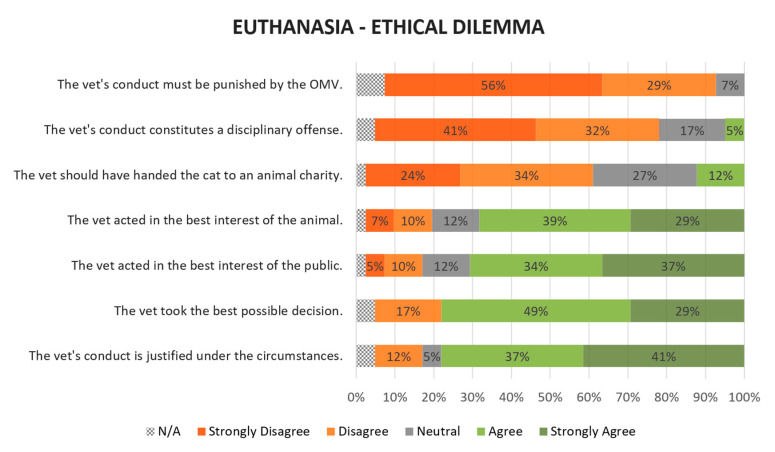
Level of agreement regarding the ethical dilemma on animal euthanasia in a shelter. Statements are listed in ascending weighted average order. Values were rounded to no decimals. On average, 4.2% of participants opted not to answer.

**Figure 4 animals-10-01596-f004:**
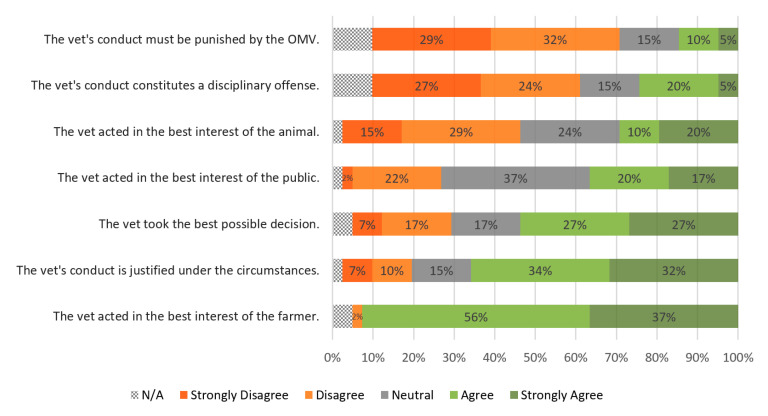
Level of agreement regarding the ethical dilemma on fitness for transport and on-farm slaughter. Statements are listed in ascending weighted average order. Values were rounded to no decimals. On average, 5.2% of participants opted not to answer.

**Figure 5 animals-10-01596-f005:**
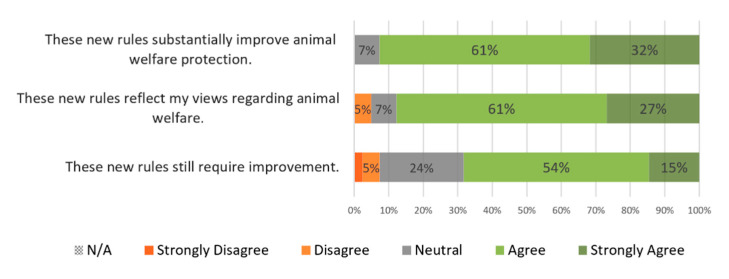
Participants’ level of agreement regarding the suggested guidance to be included in the revised Portuguese veterinary code of conduct. Statements are listed in descending weighted average order. Values were rounded to no decimals. Every participant answered all questions.

**Table 1 animals-10-01596-t001:** Vignettes used in Round 2 of the Policy Delphi, together with the research topic involved and the corresponding legal framework.

Topic	Vignette	Legal Framework
Euthanasia (animal shelter)	Marta is a local authority vet at an urban municipality in the north of Portugal. A cat that suffered a road traffic accident arrives at the animal shelter. The cat, feral and difficult to handle, has tail fractures and lacerations that require surgery. It also presents generalized dermatophytosis (ringworm). Marta assesses the case and decides to euthanase the cat. However, Marta fears that she may incur in serious disciplinary offense because it can be argued that the case falls outside the exceptions provided by the law.	Law no. 27/2016—Restricts the euthanasia of animals to “cases of proven incurable disease and when it proves to be the only means to eliminate irretrievable pain and suffering”.
Fitness for transport and on-farm slaughter	Pedro is an assistant veterinarian at a dairy farm in the district of Évora. He was called because of a downer cow, that the farmer wants to cull. In the absence of adequate means to perform on-farm emergency slaughter, Pedro opts for casualty slaughter at the slaughterhouse. The vehicle is prepared with a litter, the cow is hoisted in order to avoid trauma as much as possible, and transported to the nearest slaughterhouse (45 min drive). “It’s the best for everyone,” says Pedro. “Within an hour the animal will be slaughtered and the carcass can be salvaged. Euthanizing the animal on farm would do nothing to solve the problem and would have caused unnecessary loss”.	European Council Regulation (EC) no. 1/2005—Animals that are unable to move independently without pain or to walk unassisted shall not be considered fit for transport.

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
