# Peer review of "Establishing Animal Welfare Rules of Conduct for the Portuguese Veterinary Profession—Results from a Policy Delphi with Vignettes"

_animals, 2020, doi:10.3390/ani10091596_

Round 1
Reviewer 1 Report
This MS is informative for the prevailing veterinarians’ attitudes toward animals and their welfare as it reflects the attitudes of both the responders and the authors of this study, the latter being representative of the most proactive members (an elite) of the Portuguese veterinary community. Both groups, the researchers and their responders, reveal the familiar, traditional view of animal welfare as the lack of pain/suffering, especially in the design and responses to the injured cat vignette. However, at least one responder goes beyond that in raising the expected quality of this cat’s life, an ethical dimension that is ignored by the authors together with 21st century ethical view of mammals and other animal agents as having an inherent value of individual lives. This modern view is expressed in the 2016 Portuguese Animal Protection Act that the authors seem to call into question as contrary to the execution of “moral agency” by Portuguese vets.
What I consider a major flaw of the authors’ approach is to base ethical rules for veterinary conduct on the current prevailing views that have been shaped by anthropocentric veterinary education and, to a various extent, bias the “moral agency” of an average veterinarian. This makes an a priori assessment to this agency as something more reliable than an ethics code highly questionable – Richard Ryder said something to the effect that, with notable exceptions, the vets were conspicuously absent from all major proanimal campaigns. Not unexpectedly, in the revised guidance the respect for animals comes down to avoiding violence and unnecessary suffering, without any reference to the (positive) value of an individual life which is of critical importance for the decisions on euthanasia. The injured cat vignette does not address the chance of providing a good life for the cat after his/her recovery and thus makes an informed bioethical decision impossible – I would have joined the 4-5% minority of responders who declined to respond because of the lack of sufficient information, although my feeling is that human convenience rather than an honest utilitarian calculus prevailed in both cases (the cat should have been saved and the cow humanely killed on the spot).
Minor corrections
Both simple summary and abstract should clearly state that 41 out of 70 veterinarians responded.
Line 85 Change to: Against this background that requires...
Line 94 replace restrict by restricted
Line 113 replace inspired in real-life cases by inspired by real-life cases
Line289 “comprised” seems awkward to me in this sentence
Reviewer 2 Report
The paper addresses a very important topic and the study is interesting and useful.
a. Some general considerations:
1. Introduction:
It is proposed to better clarify the link between:
- animal welfare and animal protection
- animal welfare and moral standing of animals and ethical dilemmas
- animal welfare and issues of euthanasia and humane killing, explaining better why a special focus is dedicated to these issues
In this way 305ss lines are more clear and effective
2. A concluding paragraph to better frame the results in relation to the objectives of the study is suggested.
3. An English check in some parts is suggested
b. Below some detailed suggestions:
lines 13 and 24: I would add ‘issues like..’
line 29: the link between animal welfare and ethical dilemmas like euthanasia and emergency slaughter is to be clarified
line 32: ‘suggested’: it is suggested to clarify what is meant here in relation to line 27 ‘requirements’
line 41: to ‘protect animals’ is not equivalent to ‘promoting animal welfare’
line 46: to protect and ‘to promote’
line 48: same as line 41
lines 54-55: it is suggested to make explicit the link between animal welfare, moral duties and euthanasia and humane killing
line 66: it is suggested to make explicit why now we talk about farm animals
line 100: the Policy Delphi ‘was applied here to investigate..’
lines 100ss: it would be useful to explain more clearly the three rounds of Delphi (Figure 1 helps but lines 110ss are not very clear)
line 112: no capital letter
lines 113-114: I would not mention the reference in the text
lines 122-123: not very clear. It would be useful to detail more clearly the link between ‘supported’ and the following (see lines 256-257)
line 182: Round 2
line 275: Improve
Line 284: no capital letter
Line 286: same
Lines 287-289: ?
Line 290: changes
Line 295: with a special focus on..
Line 307: I would mention Round 2
Line 309: they were also asked to do so
Line 311: I would explain more clearly
Line 377: the link between the vignettes and the rules has to be explained and discussed more clearly
Line 382: I would mention Round 3
Line 386ss: the passage has to be explained more clearly (e.g. line 395: ethical challenges ‘when confronted with animal welfare issues’; 397 ‘involving animal welfare’; ethical decision making ‘regarding animal welfare’) The focus is on animal welfare (see Line 40)
Line 407: I would expand these considerations also in relation to Round 3
Line 408ss: A Conclusion paragraph is suggested
Line 409: and improve animal welfare
